# Efficient Formal Safety Analysis of Neural Networks

**Shiqi Wang, Kexin Pei, Justin Whitehouse, Junfeng Yang, Suman Jana**
Columbia University, NYC, NY 10027, USA
{tcwangshiqi, kpei, jaw2228, junfeng, suman}@cs.columbia.edu

## Abstract

Neural networks are increasingly deployed in real-world safety-critical domains such as autonomous driving, aircraft collision avoidance, and malware detection. However, these networks have been shown to often mispredict on inputs with minor adversarial or even accidental perturbations. Consequences of such errors can be disastrous and even potentially fatal as shown by the recent Tesla autopilot crashes. Thus, there is an urgent need for formal analysis systems that can rigorously check neural networks for violations of different safety properties such as robustness against adversarial perturbations within a certain $L$-norm of a given image. An effective safety analysis system for a neural network must be able to either ensure that a safety property is satisfied by the network or find a counterexample, i.e., an input for which the network will violate the property. Unfortunately, most existing techniques for performing such analysis struggle to scale beyond very small networks and the ones that can scale to larger networks suffer from high false positives and cannot produce concrete counterexamples in case of a property violation. In this paper, we present a new efficient approach for rigorously checking different safety properties of neural networks that significantly outperforms existing approaches by multiple orders of magnitude. Our approach can check different safety properties and find concrete counterexamples for networks that are $10\times$ larger than the ones supported by existing analysis techniques. We believe that our approach to estimating tight output bounds of a network for a given input range can also help improve the explainability of neural networks and guide the training process of more robust neural networks.

## 1 Introduction

Over the last few years, significant advances in neural networks have resulted in their increasing deployments in critical domains including healthcare, autonomous vehicles, and security. However, recent work has shown that neural networks, despite their tremendous success, often make dangerous mistakes, especially for rare corner case inputs. For example, most state-of-the-art neural networks have been shown to produce incorrect outputs for adversarial inputs specifically crafted by adding minor human-imperceptible perturbations to regular inputs [36, 14]. Similarly, seemingly minor changes in lighting or orientation of an input image have been shown to cause drastic mispredictions by the state-of-the-art neural networks [29, 30, 37]. Such mistakes can have disastrous and even potentially fatal consequences. For example, a Tesla car in autopilot mode recently caused a fatal crash as it failed to detect a white truck against a bright sky with white clouds [3].

A principled way of minimizing such mistakes is to ensure that neural networks satisfy simple safety/security properties such as the absence of adversarial inputs within a certain L-norm of a given image or the invariance of the network's predictions on the images of the same object under different lighting conditions. Ideally, given a neural network and a safety property, an automated checker should either guarantee that the property is satisfied by the network or find concrete counterexamples demonstrating violations of the safety property. The effectiveness of such automated checkers hinges on how accurately they can estimate the decision boundary of the network.

However, strict estimation of the decision boundary of a neural network with piecewise linear activation functions such as ReLU is a hard problem. While the linear pieces of each ReLU node can be partitioned into two linear constraints and efficiently check separately, the total number of linear pieces grow exponentially with the number of nodes in the network [25, 27]. Therefore, exhaustive enumeration of all combinations of these pieces for any modern network is prohibitively expensive. Similarly, sampling-based inference techniques like blackbox Monte Carlo sampling may need an enormous amount of data to generate tight accurate bounds on the decision boundary [11].

In this paper, we propose a new efficient approach for rigorously checking different safety properties of neural networks that significantly outperform existing approaches by multiple orders of magnitude. Specifically, we introduce two key techniques. First, we use *symbolic linear relaxation* that combines symbolic interval analysis and linear relaxation to compute tighter bounds on the network outputs by keeping track of relaxed dependencies across inputs during interval propagation when the actual dependencies become too complex to track. Second, we introduce a novel technique called *directed constraint refinement* to iteratively minimize the errors introduced during the relaxation process until either a safety property is satisfied or a counterexample is found. To make the refinement process efficient, we identify the potentially *overestimated nodes*, i.e., the nodes where inaccuracies introduced during relaxation can potentially affect the checking of a given safety property, and use off-the-shelf solvers to focus only on those nodes to further tighten their output ranges.

We implement our techniques as part of Neurify, a system for rigorously checking a diverse set of safety properties of neural networks $10\times$ larger than the ones that can be handled by existing techniques. We used Neurify to check six different types of safety properties of nine different networks trained on five different datasets. Our experimental results show that on average Neurify is $5,000\times$ faster than Reluplex [17] and $20\times$ than ReluVal [39].

Besides formal analysis of safety properties, we believe our method for efficiently estimating tight and rigorous output ranges of a network will also be useful for guiding the training process of robust networks [42, 32] and improving explainability of the decisions made by neural networks [34, 20, 23].

**Related work.** Several researchers have tried to extend and customize Satisfiability Modulo Theory (SMT) solvers for estimating decision boundaries with strong guarantees [17, 18, 15, 10, 31]. Another line of research has used Mixed Integer Linear Programming (MILP) solvers for such analysis [38, 12, 7]. Unfortunately, the efficiency of both of these approaches is severely limited by the high nonlinearity of the resulting formulas.

Different convex or linear relaxation techniques have also been used to strictly approximate the decision boundary of neural networks. While these techniques tend to scale significantly better than solver-based approaches, they suffer from high false positive rates and struggle to find concrete counterexamples demonstrating violations of safety properties [42, 32, 13, 9]. Similarly, existing works on finding lower bounds of adversarial perturbations to fool a neural network also suffer from the same limitations [28, 41]. Note that concurrent work of Weng et al. [40] uses similar linear relaxation method as ours but it alone struggles to solve such problems as shown in Table 6. Also, their follow-up work [44] that provides a generic relaxation method for general activation functions does not address this issue either. In contrast, we mainly use our relaxation technique to identify crucial nodes and iteratively refine output approximations over these nodes with the help of linear solver. Another line of research has focused on strengthening network robustness either by incorporating these relaxation methods into training process [43, 8, 24] or by leveraging techniques like differential privacy [22]. Our method, essentially providing a more accurate formal analysis of a network, can potentially be incorporated into training process to further improve network robustness.

Recently, ReluVal, by Wang et al. [39], has used interval arithmetic [33] for rigorously estimating a neural network's decision boundary by computing tight bounds on the outputs of a network for a given input range. While ReluVal achieved significant performance gain over the state-of-the-art solver-based methods [17] on networks with a small number of inputs, it struggled to scale to larger networks (see detailed discussions in Section 2).

## 2   Background

We build upon two prior works [10, 39] on using interval analysis and linear relaxations for analyzing neural networks. We briefly describe them and refer interested readers to [10, 39] for more details.

**Symbolic interval analysis.** Interval arithmetic [33] is a flexible and efficient way of rigorously estimating the output ranges of a function given an input range by computing and propagating the output intervals for each operation in the function. However, naive interval analysis suffers from large overestimation errors as it ignores the input dependencies during interval propagation. To minimize such errors, Wang et al. [39] used symbolic intervals to keep track of dependencies by maintaining linear equations for upper and lower bounds for each ReLU and concretizing only for those ReLUs that demonstrate non-linear behavior for the given input intervals. Specifically, consider an intermediate ReLU node $z = Relu(Eq), (l, u) = (\underline{Eq}, \overline{Eq})$, where $Eq$ denotes the symbolic representation (i.e., a closed-form equation) of the ReLU's input in terms of network inputs X and $(l, u)$ denote the concrete lower and upper bounds of $Eq$, respectively. There are three possible output intervals that the ReLU node can produce depending on the bounds of $Eq$: (1) $z = [Eq, Eq]$ when $l \geq 0$, (2) $z = [0, 0]$ when $u \leq 0$, or (3) $z = [l, u]$ when $l < 0 < u$. Wang et al. will concretize the output intervals for this node only if the third case is feasible as the output in this case cannot be represented using a single linear equation.

**Bisection of input features.** To further minimize overestimation, [39] also proposed an iterative refinement strategy involving repeated input bisection and output reunion. Consider a network $F$ taking $d$-dimensional input, and the $i$-th input feature interval is $X_i$ and network output interval is $F(X)$ where $X = \{X_1, ..., X_d\}$. A single bisection on $X_i$ will create two children: $X' = \{X_1, ..., [\underline{X_i}, \frac{X_i + \overline{X_i}}{2}], ..., X_d\}$ and $X'' = \{X_1, ..., [\frac{X_i + \overline{X_i}}{2}, \overline{X_i}], ..., X_d\}$. The reunion of the corresponding output intervals $F(X') \bigcup F(X'')$, will be tighter than the original output interval, i.e., $F(X') \bigcup F(X'') \subseteq F(X)$, as the Lipschitz continuity of the network ensures that the overestimation error decreases as the width of input interval becomes smaller. However, the efficiency of input bisection decreases drastically as the number of input dimensions increases.

**Linear relaxation.** Ehlers et al. [10] used linear relaxation of ReLU nodes to strictly over-approximate the non-linear constraints introduced by each ReLU. The generated linear constraints can then be efficiently solved using a linear solver to get bounds on the output of a neural network for a given input range. Consider the simple ReLU node taking input $z'$ with an upper and lower bound $u$ and $l$ respectively and

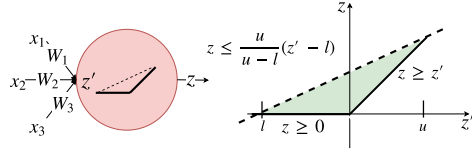

Figure 1: Linear relaxation of a ReLU node.

producing output $z$ as shown in Figure 1. Linear relaxation of such a node will use the following three linear constraints: (1) $z \geq 0$, (2) $z \geq z'$, and (3) $z \leq \frac{u(z'-l)}{u-l}$ to expand the feasible region to the green triangle from the two original piecewise linear components. The effectiveness of this approach heavily depends on how accurately $u$ and $l$ can be estimated. Unfortunately, Ehlers et al. [10] used naive interval propagation to estimate $u$ and $l$ leading to large overestimation errors. Furthermore, their approach cannot efficiently refine the estimated bounds and thus cannot benefit from increasing computing power.

## 3 Approach

In this paper, we make two major contributions to scale formal safety analysis to networks significantly larger than those evaluated in prior works [17, 10, 42, 39]. First, we combine symbolic interval analysis and linear relaxation (described in Section 2) in a novel way to create a significantly more efficient propagation method–*symbolic linear relaxation*–that can achieve substantially tighter estimations (evaluated in Section 4). Second, we present a technique for identifying the overestimated intermediate nodes, i.e., the nodes whose outputs are overestimated, during symbolic linear relaxation and propose *directed constraint refinement* to iteratively refine the output ranges of these nodes. In Section 4, we also show that this method mitigates the limitations of input bisection [39] and scales to larger networks.

Figure 2 illustrates the high-level workflow of Neurify. Neurify takes in a range of inputs $X$ and then determines using linear solver whether the output estimation generated by symbolic linear relaxation satisfies the safety proprieties. A property is proven to be safe if the solver find the relaxed constraints unsatisfiable. Otherwise, the solver returns potential counterexamples. Note that the returned counterexamples found by the solver might be false positives due to the inaccuracies

introduced by the relaxation process. Thus Neurify will check whether a counterexample is a false positive. If so, Neurify will use directed constraint refinement guided by symbolic linear relaxation to obtain a tighter output bound and recheck the property with the solver.

## 3.1 Symbolic Linear Relaxation

The symbolic linear relaxation of the output of each ReLU $z = Relu(z')$ leverages the bounds on $z'$, $Eq_{low}$ and $Eq_{up}$ ($Eq_{low} \le Eq^*(x) \le Eq_{up}$). Here $Eq^*$ denotes the closed-form representation of $z'$.

Specifically, Equation 1 shows the symbolic linear relaxation where $\mapsto$ denotes "relax to". In addition, $[l_{low}, u_{low}]$ and $[l_{up}, u_{up}]$ denote the concrete lower and upper bounds for $Eq_{low}$ and $Eq_{up}$, respectively. In supplementary material Section 1.2, we give a detailed proof showing that this relaxation is the tightest achievable due to its least maximum distance from $Eq^*$. In the following discussion, we simplify $Eq_{low}$ and $Eq_{up}$ as $Eq$ and the corresponding lower and upper bounds as $[l, u]$. Figure 3 shows the difference between our symbolic relaxation process and the naive concretizations used by Wang et al. [39]. More detailed discussions can be found in supplementary material Section 2.

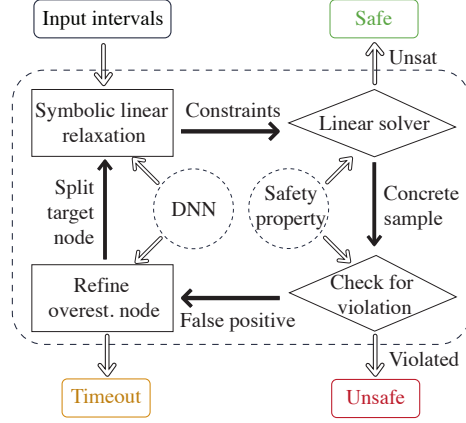

Figure 2: Workflow of Neurify to formally analyze safety properties of neural networks.

$$Relu(Eq_{low}) \mapsto \frac{u_{low}}{u_{low} - l_{low}}(Eq_{low}) \qquad Relu(Eq_{up}) \mapsto \frac{u_{up}}{u_{up} - l_{up}}(Eq_{up} - l_{up}) \qquad (1)$$

In practice, symbolic linear relaxation can cut (on average) 59.64% more overestimation error than symbolic interval analysis (cf. Section 2) and saves the time needed to prove a property by several orders of magnitude (cf. Section 4). There are three key reasons behind such significant performance improvement. First, the maximum possible error after introducing relaxations is $\frac{-l_{up}*u_{up}}{u_{up}-l_{up}}$ for upper bound and $\frac{-l_{low}*u_{low}}{u_{low}-l_{low}}$ for lower bound in Figure 3(b) (the proof is in supplementary material Section 1.2). These relaxations are considerably

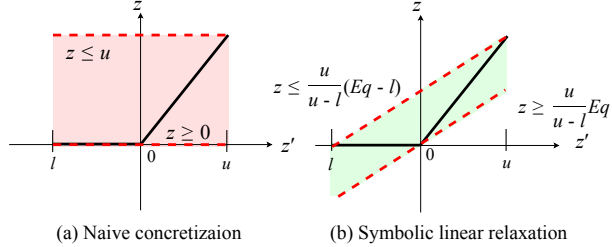

(a) Naive concretizaion    (b) Symbolic linear relaxation

Figure 3: An illustration of symbolic linear relaxation for an intermediate node. (a) Original symbolic interval analysis [39] used naive concretization. (b) Symbolic linear relaxation leverages the knowledge of concrete bounds for $z'$ and computes relaxed *symbolic interval*. $Eq$ is the symbolic representation of $z'$.

tighter than naive concretizations shown in Figure 3(a), which introduces a larger error $u_{up}$. Second, symbolic linear relaxation, unlike naive concretization, partially keeps the input dependencies during interval propagation ($[\frac{u}{u-l}Eq, \frac{u}{u-l}(Eq - l)]$) by maintaining symbolic equations. Third, as the final output error is exponential to the error introduced at each node (proved in supplementary 1.2), tighter bounds on earlier nodes produced by symbolic relaxation significantly reduce the final output error.

## 3.2 Directed Constraint Refinement

Besides symbolic linear relaxation, we also develop another generic approach, *directed constraint refinement*, to further improve the overall performance of property checking. Our empirical results in Section 4 shows the substantial improvement from using this approach combined with symbolic linear relaxation. In the following, we first define *overestimated nodes* before describing the directed constraint refinement process in detail.

**Overestimated nodes.** We note that, for most networks, only a small proportion of intermediate ReLU nodes operate in the non-linear region for a given input range $X$. These are the only nodes that

need to be relaxed (cf. Section 2). We call these nodes *overestimated* as they introduce overestimation error during relaxation. We include other useful properties and proofs regarding overestimated nodes in supplementary material Section 1.1.

Based on the definition of overestimated nodes, we define one step of directed constraint refinement as computing the refined output range $F'(X)$:

$$F'(X) = F(x \in X | Eq(x) \le 0) \cup F(x \in X | Eq(x) > 0) \tag{2}$$

where $X$ denotes the input intervals to the network, $F$ is the corresponding network, and $Eq$ is the input equation of an overestimated node. Note that here we are showing the input of a node as a single equation for simplicity instead of the upper and lower bounds shown in Section 3.1.

We iteratively refine the bounds by invoking a linear solver, allowing us to make Neurify more scalable for difficult safety properties. The convergence analysis is given in supplementary material Section 1.3.

The refinement includes the following three steps:

***Locating overestimated nodes.*** From symbolic linear relaxations, we can get the set of overestimated nodes within the network. We then prioritize the overestimated nodes with larger output gradient and refine these influential overestimated nodes first. We borrow the idea from [39] of computing the gradient of network output with respect to the input interval of the overestimated node. A larger gradient value of a node signifies that the input of that node has a greater influence towards changing the output than than the inputs of other nodes.

***Splitting.*** After locating the target overestimated node, we split its input ranges into two independent cases, $Eq_t > 0$ and $Eq_t \le 0$ where $Eq_t$ denotes the input of the target overestimated node. Now, unlike symbolic linear relaxation where $Relu([Eq_t, Eq_t]) \mapsto [\frac{u}{u-l} Eq_t, \frac{u}{u-l}(Eq_t - l)]$, neither of the two split cases requires any relaxation (Section 2) as the input interval no longer includes 0. Therefore, splitting creates two tighter approximations of the output $F(x \in X | Eq_t(x) > 0)$ and $F(x \in X | Eq_t(x) \le 0)$.

***Solving.*** We solve the resulting linear constraints, along with the constraints defined in safety properties, by instantiating an underlying linear solver. In particular, we define safety properties that check that the confidence value of a target output class $F^t$ is always greater than the outputs of other classes $F^o$ (e.g., outputs other than 7 for an image of a hand-written 7). We thus define the constraints for safety properties as $Eq_{low}^t - Eq_{up}^o < 0$. Here, $Eq_{low}^t$ and $Eq_{up}^o$ are the lower bound equations for $F^t$ and the upper bound equations for $F^o$ derived using symbolic linear relaxation. Each step of directed constraint refinement of an overestimated node results in two independent problems as shown in Equation 3 that can be checked with a linear solver.

$$\text{Check Satifiability: } Eq_{low1}^t - Eq_{up1}^o < 0; \ Eq_t \le 0; \ x_i - \epsilon \le x_i \le x_i + \epsilon \ (i = 1 \dots d)$$
$$\text{Check Satifiability: } Eq_{low2}^t - Eq_{up2}^o < 0; \ Eq_t > 0; \ x_i - \epsilon \le x_i \le x_i + \epsilon \ (i = 1 \dots d) \tag{3}$$

In this process, we invoke the solver in two ways. (1) If the solver tells that both cases are unsatisfiable, then the property is formally proved to be safe. Otherwise, further iterative refinement steps can be applied. (2) If either case is satisfiable, we treat the solutions returned by the linear solver as potential counterexamples violating the safety properties. Note that these solutions might be false positives due to the inaccuracies introduced during the relaxation process. We thus resort to directly executing the target network with the solutions returned from the solver as input. If the solution does not violate the property, we repeat the above process for another overestimated node (cf. Figure 2).

### 3.3 Safety Properties

In this work, we support checking diverse safety properties of networks including five different classes of properties based on the input constraints. Particularly, we specify the safety properties of neural network based on defining constraints on its input-output. For example, as briefly mentioned in Section 3.1, we specify that the output of the network on input $x$ should not change (i.e., remain invariant) when $x$ is allowed to vary within a certain range $X$. For output constraints, taking an arbitrary classifier as an example, we define the output invariance by specifying the difference greater than 0 between lower and upper bound of confidence value of the original class of the input and other classes. For specifying input constraints, we consider three popular bounds, i.e., $L_\infty$,

$L_1$, and $L_2$, which are widely used in the literature of adversarial machine learning [14]. These three bounds allow for arbitrary perturbations of the input features as long as the corresponding norms of the overall perturbation are within a certain threshold. In addition to these arbitrary perturbations, we consider two specific perturbations that change brightness and contrast of the input images as discussed in [30]. Properties specified using $L_\infty$ naturally fit into our symbolic linear relaxation process where each input features are bounded by an interval. For properties specified in $L_1 \leq \epsilon$ or $L_2 \leq \epsilon$, we need to add more constraints, i.e., $\sum_{i=1}^{d} |x_i| \leq \epsilon$ for $L_1$, or $\sum_{i=1}^{d} x_i^2 \leq \epsilon$ for $L_2$, which are no longer linear. We handle such cases by using solvers that support quadratic constraints (see details in Section 4). The safety properties involving changes in brightness and contrast can be efficiently checked by iteratively bisecting the input nodes simultaneously as $min_{x \in [x-\epsilon, x+\epsilon]}(F(x)) = min(min_{x \in [x, x+\epsilon]}(F(x)), min_{x \in [x-\epsilon, x]}(F(x)))$ where $F$ represents the computation performed by the target network .

# 4 Experiments

**Implementation.** We implement Neurify with about 26,000 lines of `C` code. We use the highly optimized `OpenBLAS`[1] library for matrix multiplications and lp_solve 5.5[2] for solving the linear constraints generated during the directed constraint refinement process. We further use Gurobi 8.0.0 solver for $L_2$-bounded safety properties. All our evaluations were performed on a Linux server running Ubuntu 16.04 with 8 CPU cores and 256GB memory. Besides, Neurify uses optimization like thread rebalancing for parallelization and outward rounding to avoid incorrect results due to floating point imprecision. Details of such techniques can be found in Section 3 of the supplementary material.

Table 1: Details of the evaluated networks and corresponding safety properties. The last three columns summarize the number of safety properties that are satisfied, violated, and timed out, respectively as found by Neurify with a timeout threshold of 1 hour.

| Dataset | Models | # of ReLUs | Architecture | Safety Property | Safe | Violated | Timeout |
|---|---|---|---|---|---|---|---|
| ACAS Xu [16] | ACAS Xu | 300 | <5, 50, 50, 50, 50, 50, 50, 5>[#] | C.P.* in [39] | 141 | 37 | 0 |
| MNIST [21] | MNIST_FC1 | 48 | <784, 24, 24, 10>[#] | $L_\infty$ | 267 | 233 | 0 |
| | MNIST_FC2 | 100 | <784, 50, 50, 10>[#] | $L_\infty$ | 271 | 194 | 35 |
| | MNIST_FC3 | 1024 | <784, 512, 512, 10>[#] | $L_\infty$ | 322 | 41 | 137 |
| | MNIST_CN | 4804 | <784, k:16*4*4 s:2, k:32*4*4 s:2, 100, 10>[+] | $L_\infty$ | 91 | 476 | 233 |
| Drebin [5] | Drebin_FC1 | 100 | <545334, 50, 50, 2>[#] | C.P.* in [29] | 458 | 21 | 21 |
| | Drebin_FC2 | 210 | <545334, 200, 10, 2>[#] | | 437 | 22 | 41 |
| | Drebin_FC3 | 400 | <545334, 200, 200, 2>[#] | | 297 | 27 | 176 |
| Car [2] | DAVE | 10276 | <30000, k:24*5*5 s:5, k:36*5*5 s:5, 100, 10>[+] | $L_\infty$, $L_1$, Brightness, Contrast | 80 | 82 | 58 |

\* Custom properties.
\# <$x, y, ...$> denotes hidden layers with x neurons in first layer, y neurons in second layer, etc.
\+ $k$:$c$*$w$*$h$ s:$stride$ denotes the output channel ($c$), kernel width ($w$), height ($h$) and stride ($stride$).

## 4.1 Properties Checked by Neurify for Each Model

**Summary.** To evaluate the performance of Neurify, we test it on nine models trained over five datasets for different tasks where each type of model includes multiple architectures. Specifically, we evaluate on fully connected ACAS Xu models [16], three fully connected Drebin models [5], three fully connected MNIST models [21], one convolutional MNIST model [42], and one convolutional self-driving car model [2]. Table 1 summarizes the detailed structures of these models. We include more detailed descriptions in supplementary material Section 4. All the networks closely follow the publicly-known settings and are either pre-trained or trained offline to achieve comparable performance to the real-world models on these datasets.

We also summarize the safety properties checked by Neurify in Table 1 with timeout threshold set to 3,600 seconds. Here we report the result of the self-driving care model (DAVE) to illustrate how we define the safety properties and the numbers of safe and violated properties found by Neurify. We report the other results in supplementary material Section 5.

Table 2: Different safety properties checked by Neurify out of 10 random images on Dave within 3600 seconds.

(a) $||X' - X||_\infty \leq \epsilon$

| $\epsilon$ | 1 | 2 | 5 | 8 | 10 |
|---|---|---|---|---|---|
| Safe(%) | 50 | 10 | 0 | 0 | 0 |
| Violated(%) | 0 | 20 | 70 | 100 | 100 |
| Timeout(%) | 50 | 70 | 30 | 0 | 0 |

(b) $||X' - X||_1 \leq \epsilon$

| $\epsilon$ | 100 | 200 | 300 | 500 | 700 |
|---|---|---|---|---|---|
| Safe(%) | 100 | 100 | 10 | 10 | 0 |
| Violated(%) | 0 | 0 | 40 | 50 | 60 |
| Timeout(%) | 0 | 0 | 50 | 40 | 40 |

(c) Brightness: $X - \epsilon \leq X' \leq X + \epsilon$

| $\epsilon$ | 10 | 70 | 80 | 90 | 100 |
|---|---|---|---|---|---|
| Safe(%) | 100 | 30 | 20 | 10 | 10 |
| Violated(%) | 0 | 30 | 50 | 60 | 70 |
| Timeout(%) | 0 | 40 | 30 | 30 | 20 |

(d) Contrast: $\epsilon X \leq X' \leq X$ or $X \leq X' \leq \epsilon X$

| $\epsilon$ | 0.2 | 0.5 | 0.99 | 1.01 | 2.5 |
|---|---|---|---|---|---|
| Safe(%) | 0 | 10 | 100 | 100 | 0 |
| Violated(%) | 70 | 20 | 0 | 0 | 50 |
| Timeout(%) | 30 | 70 | 0 | 0 | 50 |

**Dave.** We show that Neurify is the first formal analysis tool that can systematically check different safety properties for a large (over 10,000 ReLUs) convolutional self-driving car network, Dave [2, 6]. We use the dataset from Udacity self-driving car challenge containing 101,396 training and 5,614 testing samples [4]. Our model's architecture is similar to the DAVE-2 self-driving car architecture from NVIDIA [6, 2] and it achieves similar 1-MSE as models used in [29]. We formally analyze the network with inputs bounded by $L_\infty$, $L_1$, brightness, and contrast as described in Section 3.3. We define the safe range of deviation of the output steering direction from the original steering angle to be less than 30 degrees. The total number of cases Neurify can verify are shown in Table 2.

Table 3: Total cases that can be verified by Neurify on three Drebin models out of 100 random malware apps. The timeout setting here is 3600 seconds.

| Models | Cases(%) | 10 | 50 | 100 | 150 | 200 |
|---|---|---|---|---|---|---|
| Drebin_FC1 | Safe | 0 | 1 | 3 | 5 | 12 |
| | Violated | 100 | 98 | 97 | 86 | 77 |
| | Total | 100 | 99 | 100 | 91 | 89 |
| Drebin_FC2 | Safe | 0 | 4 | 4 | 6 | 8 |
| | Violated | 100 | 96 | 90 | 81 | 70 |
| | Total | 100 | 100 | 94 | 87 | 78 |
| Drebin_FC3 | Safe | 0 | 4 | 4 | 4 | 15 |
| | Violated | 100 | 89 | 74 | 23 | 11 |
| | Total | 100 | 93 | 78 | 33 | 26 |

**DREBIN.** We also evaluate Neurify on three different Drebin models containing 545,334 input features. The safety property we check is that simply adding app permissions without changing any functionality will not cause the models to misclassify malware apps as benign. Here we show in Table 3 that Neurify can formally verify safe and unsafe cases for most of the apps within 3,600 seconds.

## 4.2 Comparisons with Other Formal Checkers

**ACAS Xu.** Unmanned aircraft alert systems (ACAS Xu) [19] are networks advising steering decisions for aircrafts, which is on schedule to be installed in over 30,000 passengers and cargo aircraft worldwide [26] and US Navy's fleets [1]. It is comparably small and only has five input features so that ReluVal [39] can efficiently check different safety properties. However, its performance still suffers from the over-approximation of output ranges due to the concretizations introduced during symbolic interval analysis. Neurify leverages symbolic linear relaxation and achieves on average $20\times$ better performance than ReluVal [39] and up to $5,000\times$ better performance than Reluplex [17]. In Table 4, we summarize the time and speedup of Neurify compared to ReluVal and Reluplex for all the properties tested in [17, 39].

Table 4: Performance comparisons of Neurify, Reluplex, and ReluVal while checking different safety properties of ACAS Xu. $\phi_1$ to $\phi_{10}$ are the properties tested in [17]. $\phi_{11}$ to $\phi_{15}$ are the additional properties tested in [39].

| Source | Props | Reluplex (sec) | ReluVal (sec) | Neurify (sec) | $\frac{Reluplex}{Neurify}$ ($\times$) | $\frac{ReluVal}{Neurify}$ ($\times$) |
|---|---|---|---|---|---|---|
| Security Properties from [17] | $\phi_1$ | >443,560.73* | 14,603.27 | 458.75 | $> 967\times$ | $31.83\times$ |
| | $\phi_2^{**}$ | 123,420.40 | 117,243.26 | 16491.83 | $>8\times$ | $7.11\times$ |
| | $\phi_3$ | 35,040.28 | 19,018.90 | 600.64 | $58.33\times$ | $31.66\times$ |
| | $\phi_4$ | 13,919.51 | 441.97 | 54.56 | $255\times$ | $8.10\times$ |
| | $\phi_5$ | 23,212.52 | 216.88 | 21.378 | $1086\times$ | $10.15\times$ |
| | $\phi_6$ | 220,330.82 | 46.59 | 1.48 | $148872\times$ | $31.48\times$ |
| | $\phi_7$ | >86400.00* | 9,240.29 | 563.55 | $>154\times$ | $16.40\times$ |
| | $\phi_8$ | 43,200.01 | 40.41 | 33.17 | $1302\times$ | $1.22\times$ |
| | $\phi_9$ | 116,441.97 | 15,639.52 | 921.06 | $126.42\times$ | $16.98\times$ |
| | $\phi_{10}$ | 23,683.07 | 10.94 | 1.16 | $20416.38\times$ | $9.43\times$ |
| Additional Security Properties | $\phi_{11}$ | 4,394.91 | 27.89 | 0.62 | $7089\times$ | $44.98\times$ |
| | $\phi_{12}$ | 2,556.28 | 0.104 | 0.13 | $19664\times$ | $0.80\times$ |
| | $\phi_{13}$ | >172,800.00* | 148.21 | 38.11 | $>4534\times$ | $3.89\times$ |
| | $\phi_{14}$ | >172,810.86* | 288.98 | 22.87 | $>7556\times$ | $12.64\times$ |
| | $\phi_{15}$ | 31,328.26 | 876.8 | 91.71 | $342\times$ | $9.56\times$ |

\* Reluplex uses different timeout thresholds for different properties.
\*\* Reluplex returns spurious counterexamples on two safe networks due to a rounding bug and ends prematurely.

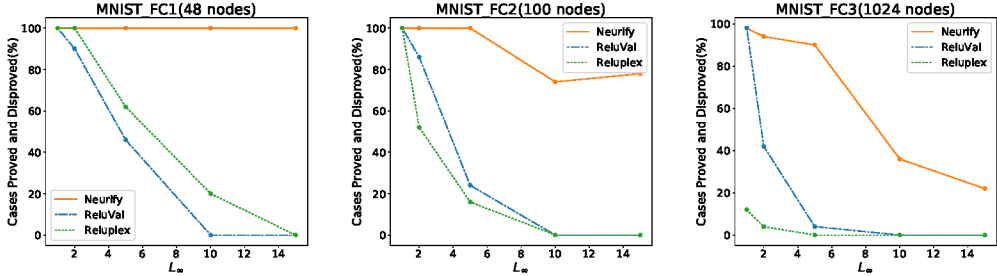

Figure 4: As we increase the $L_\infty$ bounds of the safety properties, the number of cases ReluVal and Reluplex can verify quickly decreases while Neurify clearly outperforms both of them. We use 50 randomly selected imaged for each property and set the timeout to 1,200 seconds.

**MNIST_FC.** The MNIST networks have significantly more inputs than ACAS Xu. It has 784 input features and ReluVal always times out when the analyzed input ranges become larger ($L_\infty \geq 5$). We measure the performance of Neurify on fully connected MNIST models MNIST_FC1, MNIST_FC2 and MNIST_FC3 and compare the cases that can be verified to be safe or a counterexample can be found by ReluVal and Reluplex in Figure 4. The results show that Neurify constantly outperforms the other two. Especially when increasing the $L_\infty$ bound, the percentages of properties that the other two can verify quickly decrease. Note that the increase in Figure 4b is caused by the more unsafe cases detected by Neurify. Initially, when the bounds are small, Neurify can easily check the properties to be safe. But as the bounds get larger, the number of verified safe cases drop drastically because (i) the underlying model tends to have real violations and (ii) Neurify suffers from relatively higher overestimation errors. However, as the bounds increase further, the counterexamples become frequent enough to be easily found by Neurify. Therefore, such phenomenon indicates that Neurify can find counterexamples more effectively than ReluVal and Reluplex due to its tighter approximation.

### 4.3 Benefits of Each Technique

**Symbolic Linear Relaxation (SLR).** We compare the widths of estimated output ranges computed by naive interval arithmetic [39] and symbolic linear relaxation on MNIST_FC1, MNIST_FC2, MNIST_FC3. We summarize the average output widths in Table 5. The experiments are based on 100 images each bounded

Table 5: The average widths of output ranges of three MNIST models for 100 random images where each has five different $L_\infty \leq \{1, 5, 10, 15, 25\}$.

| | NIA* | SLR** | Improve(%) |
|---|---|---|---|
| MNIST_FC1 | 111.87 | 52.22 | 114.23 |
| MNIST_FC2 | 230.27 | 101.72 | 126.38 |
| MNIST_FC3 | 1271.19 | 624.27 | 103.63 |

\* Naive Interval Arithmetic
\*\* Symbolic Linear Relaxation

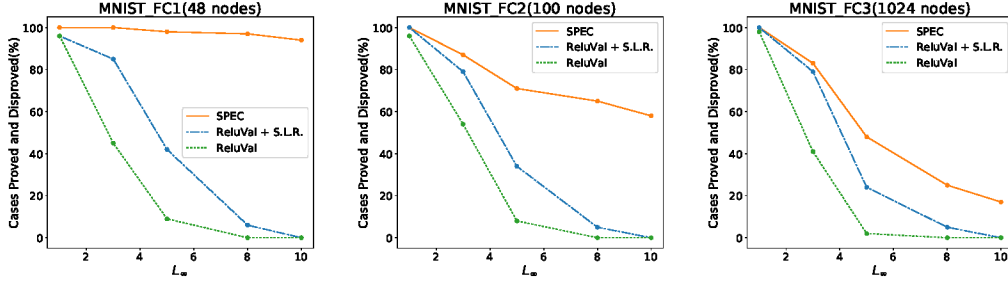

Figure 5: Showing the cases out of 100 randomly selected images that can be verified by Neurify, new ReluVal+SLR and original ReluVal. Here ReluVal+SLR denotes the new ReluVal improved with our symbolic linear relaxation for showing the performance of DCR. The timeout setting is 600 seconds.

by $L_\infty \leq \epsilon$ ($\epsilon = 1, ..., 25$). The results indicate that SLR in Neurify can tighten the output intervals by at least 100% over naive interval arithmetic, which significantly speeds up its performance.

**Directed Constraint Refinement (DCR).** To illustrate how DCR can improve the overall performance when combined with SLR, we evaluate Neurify on MNIST_CN and measure the number of timeout cases out of 100 randomly selected input images when using symbolic linear relaxation alone and combining it with directed constraint refinement. Table 6 shows that SLR combined with DCR can verify 18.88% more cases on average than those using SLR alone.

**Improvements Evaluated on MNIST.** We evaluate how symbolic linear relaxation and directed constraint refinement each can improve the performance compared with Relu-Val on three fully connected MNIST models, MNIST_FC1, MNIST_FC2, and MNIST_FC3. For measuring the improvement made by symbolic linear relaxation, we integrate it into Relu-

Table 6: The timeout cases out of 100 random images generated while using symbolic linear relaxation alone and together with directed constraint refinement. These results are computed for MNIST_CN model with $L_\infty \leq \epsilon$ ($\epsilon = 1, ..., 25$). The last column shows the number of additional cases checked while using directed constraint refinement.

| Properties | SLR | SLR + DCR | Improve |
|---|---|---|---|
| $\epsilon = 1$ | 0 | 0 | 0 |
| $\epsilon = 2$ | 2 | 1 | 1 |
| $\epsilon = 3$ | 6 | 0 | 6 |
| $\epsilon = 4$ | 18 | 5 | 13 |
| $\epsilon = 5$ | 58 | 12 | 46 |
| $\epsilon = 10$ | 100 | 90 | 10 |
| $\epsilon = 15$ | 100 | 80 | 20 |
| $\epsilon = 25$ | 100 | 45 | 55 |

Val denoted as ReluVal+SLR (input bisection + symbolic linear relaxation) and we compare with the number of original ReluVal (input bisection + symbolic interval analysis). As for the performance of directed constraint refinement, we make comparisons between ReluVal+SLR (input bisection + symbolic linear relaxation) and Neurify (directed constraint refinement + symbolic linear relaxation). We summarize the total cases that can be verified by Neurify, original ReluVal, and ReluVal+SLR out of 100 random MNIST images within 600 seconds in Figure 5. The safety properties are defined as whether the models will misclassify the images within allowable perturbed input ranges bounded by $L_\infty \leq \epsilon (\epsilon = 1, ..., 10)$. The experimental results demonstrate that our symbolic linear approximation can help ReluVal find 15% more cases on average. However, the input bisection used by ReluVal+SLR still suffers from larger number of input features and thus usually times out when $\epsilon$ is large. Neurify's DCR approach mitigated that problem and additionally verify up to 65% more cases on average compared to ReluVal.

## 5 Conclusion

We designed and implemented Neurify, an efficient and scalable platform for verifying safety properties of real-world neural networks and providing concrete counterexamples. We propose symbolic linear relaxation to compute a tight over-approximation of a network's output for a given input range and use directed constraint refinement to further refine the bounds using linear solvers. Our extensive empirical results demonstrate that Neurify outperforms state-of-the-art formal analysis systems by several orders of magnitude and can easily scale to networks with more than 10,000 ReLU nodes.

# 6 Acknowledgements

We thank the anonymous reviewers for their constructive and valuable feedback. This work is sponsored in part by NSF grants CNS-16-17670, CNS-15-63843, and CNS-15-64055; ONR grants N00014-17-1-2010, N00014-16-1- 2263, and N00014-17-1-2788; and a Google Faculty Fellowship. Any opinions, findings, conclusions, or recommendations expressed herein are those of the authors, and do not necessarily reflect those of the US Government, ONR, or NSF.

## Footnotes

[1]https://www.openblas.net/

[2]http://lpsolve.sourceforge.net/5.5/

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
