[Supplementary Material]

# Efficient Formal Safety Analysis of Neural Networks Supplementary Material

This supplementary material contains the details left out of the original submission for brevity.

## 1 Proofs

### 1.1 Properties of Overestimated Nodes

Below we describe and prove some useful properties that overestimated nodes satisfy. Throughout this section, $X$ denotes an input interval range, $z = Relu(Eq)$ denotes an overestimated node, and $W$ denotes a set of overestimated nodes. Furthermore, $[l, u]$ and $[Eq_{low}, Eq_{up}]$ represent the concrete and symbolic intervals for each node before ReLU function, respectively. Lastly, we let $Eq^*$ be its ground-true equation.

**property 1.1.** *Given input range $X$, an overestimated node's ($z = Relu(Eq)$) concrete upper and lower bounds satisfy $u = max_{x \in X} Eq(x) > 0$ and $l = min_{x \in X} Eq(x) < 0$.*

**proof:** It suffices to show that $\exists x_1, x_2 \in X$ such that $Eq(x_1) > 0$ and $Eq(x_2) < 0$. If $Eq$ are strictly non-negative on $X$, then, for any $x_1 \in X$, we have $Relu(Eq(x_1)) = Eq(x_1)$ and we do not perform any relaxations. Likewise, if $Eq$ are strictly non-positive, then for any $x_2 \in X$, we have $Relu(Eq(x_2)) = 0$, so we also do not need to apply any relaxations. But, we assumed that the node was overestimated, so that there must be $\exists x_1 \in X$ such that $Eq(x_1) > 0$, and $\exists x_2 \in X$ such that $Eq(x_2) < 0$. Therefore, since $u = max_{x \in X} Eq(x) \geq Eq(x_1)$, and $l = min_{x \in X} Eq(x) \leq Eq(x_2)$, we have $l < 0$ and $u > 0$, which is the desired result.

**property 1.2.** *The symbolic input interval $[Eq_{low}, Eq_{up}]$ to a node of the $i$-th layer satisfies $Eq_{low} = Eq_{up} = Eq^*$ if there are no overestimated node in the earlier layers.*

**proof:** We prove this inductively over the number of layers in a network. For the base case, we consider the first layer, which is the input layer. The assumption that there is no overestimated node in all previous layers, in this case, is always true, as we define overestimated nodes to occur only when we apply the activation function ReLU. Thus, we have $Eq_{low} = Eq^* = Eq_{up}$ in this case. Now, suppose that the property holds for inputs up to the $i$-th layer. We show it consequently holds for inputs to the $(i + 1)$-th layer. We know, for the $j$-th node in the $i$-th layer, that its input is $[Eq_j^*, Eq_j^*]$. Since we now assume that no nodes in the $i$-th layer are overestimated nodes, we know that $Relu(Eq_j^*) = Eq_j^*$ or 0. Let $y_j$ denote its output and let $y$ denote the vector of outputs in this layer. Then, the output of the $j$-th node is $[y_j, y_j]$. Considering the weights $W$, one can see that the input for the $k$-th node of the $(i + 1)$-th layer is $[Eq_{low}, Eq_{up}] = [(Wy)_k, (Wy)_k]$. Thus, we have $Eq_{low} = Eq_{up} = Eq^*$ which is exactly the same as the claim.

**corollary 1.1.** *In a neural network that contains no overestimated node, there is no error in the output layer.*

**proof:** This follows from Property 1.2, as it tells us that, for each node, say node $j$, in the output layer, $Eq_{low} = Eq^* = Eq_{up}$. Since this holds for all nodes, there is zero error.

## 1.2 Symbolic Linear Relaxation

**lemma 1.1.** *The maximum distances between the approximation given in symbolic linear relaxation as Equation 1 are $\frac{-u_{up}l_{up}}{u_{up}-l_{up}}$ for upper bound and $\frac{-u_{low}l_{low}}{u_{low}-l_{low}}$ for lower bound.*

$$Relu(Eq_{low}) \mapsto \frac{u_{low}}{u_{low}-l_{low}}(Eq_{low}) \qquad Relu(Eq_{up}) \mapsto \frac{u_{up}}{u_{up}-l_{up}}(Eq_{up}-l_{up}) \qquad (1)$$

**proof:** The distance for upper bound in Equation 1 is:

$$
\begin{aligned}
d_{up} &= \frac{u_{up}}{u_{up}-l_{up}}(Eq_{up}-l_{up}) - Relu(Eq_{up}) \\
&= \begin{cases} \frac{u_{up}}{u_{up}-l_{up}}(Eq_{up}-l_{up}) - Eq_{up} & (if \quad 0 \le Eq_{up} \le u_{up}) \\ \frac{u_{up}}{u_{up}-l_{up}}(Eq_{up}-l_{up}) & (if \quad l_{up} \le Eq_{up} < 0) \end{cases} \\
&\le \frac{-u_{up}l_{up}}{u_{up}-l_{up}} \qquad (when \quad Eq_{up} = 0)
\end{aligned}
\qquad (2)
$$

The distance for lower bound in Equation 1 is:

$$
\begin{aligned}
d_{low} &= Relu(Eq_{low}) - \frac{u_{low}}{u_{low}-l_{low}}(Eq_{low}) \\
&= \begin{cases} Eq_{low} - \frac{u_{low}}{u_{low}-l_{low}}(Eq_{low}) & (if \quad 0 \le Eq_{low} \le u_{low}) \\ -\frac{u_{low}}{u_{low}-l_{low}}(Eq_{low}) & (if \quad l_{low} \le Eq_{low} < 0) \end{cases} \\
&\le -\frac{u_{low}l_{low}}{u_{low}-l_{low}} \qquad (when \quad Eq_{low} = l_{low}/Eq_{low} = u_{low})
\end{aligned}
\qquad (3)
$$

**property 1.3.** *The approximation produced by symbolic linear relaxation $Eq_{up}$ and $Eq_{low}$ as Equation 1 has the least maximum distance from the actual output $Eq^*$.*

**proof:** We give the proof for upper and lower symbolic linear relaxation respectively.

*(1) Upper symbolic linear relaxation:* The maximum distance for upper bound in Equation 1 is $m = \frac{-ul}{u-l}$ when $Eq(x) = 0$ shown in Lemma 1.1. If another symbolic linear relaxation that has maximum distance $m' < m$, then it can be written as $Relu(Eq_{up}) \mapsto k(Eq_{up}(x)) + m'$ due to its linearity. To overestimate two points $Relu(u) \mapsto k \cdot u + m' \ge u$ and $Relu(l) \mapsto k \cdot l \ge 0$, we arrive at the inequality $\frac{u-m'}{u} \le k \le \frac{-m'}{l}$. Consequently, we get $m' \ge \frac{-ul}{u-l}$, which conflicts with assumption $m' < m$.

*(2) Lower symbolic linear relaxation:* Also shown in Lemma 1.1, the maximum distance $m = \frac{-ul}{u-l}$ for lower bound equation can be achieved when $Eq = l$ or $Eq = u$. Assume there is another lower symbolic linear relaxation has the maximum distance $m' < m$, it can be similarly written as $Relu(Eq(x)) \mapsto \frac{u+m'_1-m'_2}{u-l}x - \frac{ul+um'_1-lm'_2}{u-l}$, where $Relu(l) \mapsto m'_1 < m$ and $Relu(u) \mapsto m'_2 < m$. To ensure $Relu(0) \mapsto -\frac{ul+um'_1-lm'_2}{u-l} \le 0$, we see $ul+um'_1-lm'_2 \ge 0$, which conflicts with $m'_1 < m$ and $m'_2 < m$.

Thus, we have shown the claim.

## 1.3 Directed Constraint Refinement

**lemma 1.2.** *If there are $n$ overestimated nodes in a neural network, then after applying directed constraint refinement to each of the $n$ nodes, that is, considering $2^n$ cases after splitting, we achieve a function $F'$ satisfying $F' = F^*$, where $F^*$ is the actual function.*

**proof:** After splitting all the $n$ overestimated nodes, for each split cases, all the nodes are constrained to be linear and thus there is no overestimated node. According to Corollary 1.1, we can see there is no error in the output layer for each case. The output union $F'$ of these $2^n$ cases is an approximation of the network without any overestimation error, which is exactly the same as the actual function $F^*$.

## 2 Detailed Study of Symbolic Linear Relaxation

As we have shown before in Section 3 of the paper, the insight of symbolic linear relaxation is in finding the tightest possible linear bounds of the ReLU function and therefore minimizing the overestimation error while approximating network outputs. Note that overestimated nodes in different layers will have different approximation errors depending on the symbolic intervals that they were given as input. For layers before $n_0$-th layer, the one in which the first overestimated nodes occur, symbolic lower and upper bounds will be the same for all nodes. This is shown in Property 1.2, and makes the approximations in earlier layers a straightforward computation. However, the expressions of lower and upper symbolic bounds can be much more complicated in deeper layers where the presence of overestimated nodes becomes more frequent. To address this problem, we discuss and illustrate how symbolic linear relaxation works in detail below.

We consider an arbitrary overestimated node A, with equation given by $z = Relu(Eq)$, where its input $Eq$ is kept as a symbolic interval $[Eq_{low}, Eq_{up}]$. Furthermore, we let $n_0$ denote the first layer in which overestimated nodes occur, let $n_A$ denote the layer overestimated node $A$ appears in, and let $(l_{low}, u_{low})$ and $(l_{up}, u_{up})$ denote the concrete lower and upper bound of $A$'s symbolic bounds $Eq_{low}$ and $Eq_{up}$. We consider several cases for the symbolic linear relaxations on $A$, depending on where it appears in relation to $n_0$.

**a.** $n_A = n_0$ : If A is an overestimated node in $n_0$, then, according to Property 1.2, we see that $A$'s input equation $Eq$ satisfies $Eq_{low} = Eq_{up} = Eq$. Furthermore, due to Property 1.1, $u = max_{x \in X} Eq(x) > 0$ and $l = min_{x \in X} Eq(x) < 0$, $A's$ output can be easily approximated by:

$$Relu([Eq, Eq]) \mapsto [\frac{u}{u-l}Eq, \frac{u}{u-l}(Eq-l)] \tag{4}$$

**b.** $n_A > n_0$ : If A is an overestimated node after $n_0$-th layer, possibly its symbolic lower bound equation $Eq_{low}$ is no longer the same as its upper bound equation $Eq_{up}$ before relaxation. Though we can still approximate the it as $Relu([Eq_{low}, Eq_{up}]) \mapsto [\frac{u_{up}}{u_{up}-l_{low}}Eq_{low}, \frac{u_{up}}{u_{up}-l_{low}}(Eq_{up}-l_{low})]$, this is not the tightest possible bound. Therefore, we consider bounds on $Eq_{low}$ and $Eq_{up}$ independently to achieve tighter approximations. In details, this process can be divided into following four scenarios shown in Equation 5, each dependent on different value taken by $Eq_{low}$ and $Eq_{up}$.

$$Relu([Eq_{low}, Eq_{up}]) \mapsto$$
$$\begin{cases} [0, \frac{u_{up}}{u_{up}-l_{up}}(Eq_{up}-l_{up})] & (l_{low} \leq 0, u_{low} \leq 0, l_{up} \leq 0, u_{up} > 0) \\ [0, Eq_{up}] & (l_{low} \leq 0, u_{low} \leq 0, l_{up} > 0, u_{up} > 0) \\ [\frac{u_{low}}{u_{low}-l_{low}}Eq_{low}, \frac{u_{up}}{u_{up}-l_{up}}(Eq_{up}-l_{up})] & (l_{low} \leq 0, u_{low} > 0, l_{up} \leq 0, u_{up} > 0) \\ [\frac{u_{low}}{u_{low}-l_{low}}Eq_{low}, Eq_{up}] & (l_{low} \leq 0, u_{low} > 0, l_{up} > 0, u_{up} > 0) \end{cases}$$
$$\tag{5}$$

For instance, consider an overestimated node satisfying the third case that both of $Eq_{low}$ and $Eq_{up}$ can take concrete value spanning 0. The maximum error introduced by relaxation according to Equation 4 is $\frac{-u_{up}l_{low}}{u_{up}-l_{low}}$, while the relaxation by Equation 5 has smaller maximum error $max(\frac{-u_{up}l_{up}}{u_{up}-l_{up}}, \frac{-u_{low}l_{low}}{u_{low}-l_{low}})$. Thus, we can see such case work allows us to have tighter the approximations.

## 3 Different Optimization and Implementation Details

In our initial experiments, we found out that the performance of matrix multiplications is a major determining factor for the overall performance of the symbolic relaxation and interval propagation process. We use the highly optimized `OpenBLAS`[1] library for matrix multiplications. For solving the linear constraints generated during the directed constraint refinement process, we use `lp_solve` `5.5`[2]. For formally checking non-existence of adversarial images that can be generated by an L-2 norm bounded attacker, it requires us to solve an optimization problem where the constraints are

linear but the objective is quadratic. Since lp_solve does not support quadratic objectives yet, `Gurobi 8.0.0`[3] solver can be further used to handle these attacks..

***Parallelization.*** Our directed constraint refinement process is highly parallelizable as it creates an independent set of linear programs that can be solved in parallel. For facilitating this process, Neurify creates a thread pool where each thread solves one set of linear constraints with its own lp_solve instances. However, as the refinement process might be highly uneven for different overestimated nodes, we periodically rebalance the queues of different threads to minimize idle CPU time.

***Outward Rounding.*** One of the side effects of floating point computations is that even minor precision drops on one hidden node can be amplified significantly during propagations. To avoid such issues, we perform outward rounding after every floating point computation, i.e., we always round $[\underline{x}, \overline{x}]$ to $[\lfloor \underline{x} \rfloor, \lceil \overline{x} \rceil]$. Our current prototype uses 32-bit `float` arithmetic that can support all of our current safety properties with outward rounding. If needed, the analysis can be easily switched to 64-bit `double`.

***Supporting Convolutional Layers.*** Models with convolutional layers are often used in computer vision applications. They usually perform matrix multiplications with a convolution kernel as shown in the dash boxes of Figure 1. To allow a symbolic interval to propagate through various convolutional layers, we simply multiply the symbolic interval inputs with the concrete kernels as shown in Figure 1.

Figure 1: Element-wise matrix multiplications to allow symbolic intervals to propagate through convolutional kernels.

# 4   Experimental Setup

To evaluate the performance of Neurify, we tested it with 9 different models, including fully connected ACAS Xu models, three fully connected Drebin models, three fully connected MNIST models, one convolutional MNIST model and one convolutional self-driving car model. The detailed structures of all these models are summarized in Manuscript Table 1 and here we provide the detailed descriptions of each type of model.

***ACAS Xu.*** ACAS is crucial aircraft alert systems used for alerting and preventing aircraft collisions. Its unmanned system ACAS Xu [8] are networks advising decisions for aircraft based on the conditions of intruders and ownships. Due to its powerful abilities, it is on schedule to be installed in over 30,000 passenger and cargo aircraft worldwide [10] and US Navy's fleets [1]. ACAS Xu is made up of 45 different models, each has five inputs, five outputs, six fully connected layers, fifty ReLU nodes in each layer. All the ACAS Xu models and self-defined safety properties we tested are given in [7, 6, 12].

***MNIST_FC.*** MNIST [9] is a handwritten digit dataset containing 28x28 pixel images with class labels from 0 to 9. The dataset includes 60,000 training samples and 10,000 testing samples. Here we use three different architectures of fully connected MNIST models with accuracies of 96.59%, 97.43%, and 98.27%. The more ReLU nodes the model has, the higher its accuracy is.

***Drebin_FC.*** Drebin [4] is a dataset with 129,013 Android applications among which 123,453 are benign and 5,560 are malicious. Currently, there is a total of 784,544 binary features extracted from each application according to 8 predefined categories [11]. The accuracy for Drebin_FC1, Drebin_FC2 and Drebin_FC3 are 97.61%, 98.53% and 99.01%. We show that compared to input bisection and refinement, directed constraint refinement can be applied on more generalized networks, such as Drebin model with such large amount of input features.

***ConvNet.*** ConvNet is a comparably large convolutional MNIST models with about 5000 ReLU nodes used in [13] trained on 60,000 MNIST dataset. Due to its large amounts of ReLU nodes, its safety property can never be supported by the traditional solver-based formal analysis systems such as ReluPlex [7].

***Self-driving Car.*** Finally, we use a large-scale (with over 10,000 ReLU nodes) convolutional autonomous vehicle model, on which no formal proof has been given before. The self-driving car

dataset comes from Udacity self-driving car challenge containing 101,396 training and 5,614 testing samples [3]. And we use similar DAVE-2 self-driving car architecture from NVIDIA [5, 2] with $3 \times 100 \times 100$ inputs and 1 regression output for advisory direction. The detailed structure is shown in Manuscript Table 1. On such model, we have already shown that Neurify can formally prove four different types of safety properties: $L_\infty$, $L_1$, Brightness and contrast on DAVE in Table 2.

# 5 Additional Results

Neurify has either formally verified or provided counterexamples for thousands of safety properties, and we summarize the additional primary properties Neurify can provide on different models in the first subsection in details. Note that the results of DAVE can be found in Manuscript Section 4.

## 5.1 Cases verified by Neurify for Each Model

**ConvNet.** We evaluate Neurify on ConvNet, a large convolutional MNIST model. To the best of our knowledge, none of traditional solver-based formal analysis systems can give formal guarantee for such large convolutional MNIST network. However, Neurify is able to verify most of the properties within $L_\infty \leq 5$. We define the safety property as whether the model will misclassify MNIST images within allowable input ranges bounded by $L_\infty \leq \epsilon (\epsilon = 1, ..., 25)$. The results are shown in Table 1. The timeout setting here is 3600 seconds.

Table 1: Showing the percentage of images formally verified (safe and violated out of 100 randomly selected images) by Neurify with different safety properties ($L_\infty \leq \epsilon (\epsilon = 1, ..., 25)$) on MNIST ConvNet.

| $\epsilon$ | Safe(%) | Violated(%) | Total(%) |
|---|---|---|---|
| 1 | 100 | 0 | 100 |
| 2 | 98 | 1 | 99 |
| 3 | 98 | 2 | 100 |
| 4 | 93 | 2 | 95 |
| 5 | 86 | 2 | 88 |
| 10 | 1 | 9 | 10 |
| 15 | 0 | 20 | 20 |
| 25 | 0 | 55 | 55 |

**MNIST_FC.** We also evaluate Neurify on three different fully connected MNIST models. The property is defined as whether the image will be misclassified with allowable perturbed input ranges bounded by $L_\infty \leq \epsilon (\epsilon = 1, ..., 15)$. In Table 2, we show the cases that Neurify can formally verify to be safe or find concrete counterexamples on out of 100 random images from MNIST dataset within 3600 seconds.

Table 2: Total cases that can be verified by Neurify on three fully connected MNIST models out of 100 hand-written digits. The timeout setting here is 3600 seconds.

| Models | Cases (%) | 1 | 2 | 5 | 10 | 15 |
|---|---|---|---|---|---|---|
| | Violated | 15 | 27 | 29 | 66 | 96 |
| MNIST_FC1 | Safe | 85 | 73 | 71 | 34 | 4 |
| | Total | 100 | 100 | 100 | 100 | 100 |
| | Violated | 0 | 11 | 26 | 74 | 83 |
| MNIST_FC2 | Safe | 100 | 89 | 68 | 9 | 5 |
| | Total | 100 | 100 | 94 | 83 | 88 |
| | Violated | 0 | 4 | 8 | 6 | 23 |
| MNIST_FC3 | Safe | 96 | 87 | 79 | 33 | 27 |
| | Total | 96 | 91 | 87 | 39 | 50 |

Figure 2: The relationship between formal analysis time and the corresponding average and maximal refinement depth for 37 ACAS Xu and 60 MNIST safety properties formally verified to be safe by Neurify.

## 5.2 Benefits of Each Technique

We have described the benefits of our two main techniques, symbolic linear relaxation (SLR) and directed constraint refinement (DCR) in Section 4 of the paper. Here, we describe additional experimental results for illustrating the benefits of each technique used in Neurify.

**Benefits of Depths in Refinement.** The process of directed constraint refinement is a DFS search tree and each iteration of refinement will generate two subtrees and thus increase one depth of whole DFS tree. We summarize the formal analysis time on 37 ACAS Xu cases and 60 MNIST cases in terms of average and maximal depth of the directed constraint refinement search tree in Figure 2. The results indicate that formal analysis time is exponential to maximum and average depth. Therefore, in practice, we can leverage depths to estimate the whole formal analysis progress.

Figure 3: Neurify's ability to locate concrete counterexamples compared to CW attacks with 5, 10, 20 random input seeds. We evaluate on MNIST_FC1, MNIST_FC2, MNIST_FC3, with 60 different safety properties that have been verified to be violated by Neurify.

**Benefits of Adversarial Searching Ability.** We show that Neurify has stronger ability to find counterexamples compared with current state-of-the-art gradient-based attack CW. Out of 60 violated cases found by Neurify on MNIST_FC1, MNIST_FC2, MNIST_FC3, CW attacks can only locate 47%, 46% and 43% with 20, 10 and 5 random input seeds, shown in Figure 3.

## Footnotes

[1] https://www.openblas.net/

[2] http://lpsolve.sourceforge.net/5.5/

[3]http://www.gurobi.com/