[Reviews · NeurIPS 2018]

Reviewer 1



The paper improves on previous methods for formal verification of neural network properties over small intervals, such as formally verifying resistance to Linf adversarial examples in a ball around one point at a time. The paper is a significant (~1 OOM) improvement relative to previous work on this problem. I am somewhat unconvinced of the importance of this type of verification given that (1) these techniques provide absolute confidence about a tiny sliver of the possible attack space, (2) as-is they are many orders of magnitude too expensive to use at training time, and therefore not much of an improvement given that most networks do not satisfy even the safety properties these techniques can check, and (3) have little to no hope of scaling to interestingly sized networks such as ImageNet models. However, since this paper is an order of magnitude speed improvement relative to previous work, I am marginally in favor of setting those concerns aside and accepting it on speed merits. The paper proceeds by maintaining two types of bounds on every value in a relu network: a concrete interval bound and a linear interval bound whose lower and upper components are linear functions in terms of other values in the network. The main novelty relative to previous work is in transforming these linear interval bounds through relu activation functions by bracketing the relu function between two linear functions over the interval given by the concrete bounds. The details of the paper appear novel relative to previous work and are relatively simple (a plus) given the OOM speedup. High level comments: 1. I would like at least a brief discussion of the limits of Linf balls as a threat model. Even restricting to human-imperceptible adversarial examples, Linf balls make up only a tiny sliver of possible image edits, and it seems unlikely that these techniques would extend to all of them. I am worried about overfitting this type of verification research to a part of the space that attackers can easily ignore. 2. Is there any hope of scaling these techniques to extremely large neural networks, such as those with more than 10k neurons? 3. I find the notation in the paper confusing, and this bleeds over into lack of clarity in the algorithm. In particular, the paper never explicitly states what type of mathematical object the Eq's are. Are they linear functions over the input? Linear functions over other nodes? I believe I can infer from the rest of the paper that it's the latter, but this should be stated. This confusion is a consequence of taking the notation from [28] and applying it without change, which doesn't work: [28] uses a purely bottom up computation but this paper solves for constraint consequences recursively using an LP solver. Relatedly, in many places in the paper the language seems to imply that [l,u] are computed from [Eq\_low, Eq\_up], which can't be true if the Eq's are just linear equations that do not know about the concrete bounds of other nodes. 4. The paper does not clearly specify how [l,u] are computed and refined. Before DCR all the [l,u]'s and Eq's are computed in a bottom up pass using interval arithmetic. Once we have a bunch of Eq < 0 or Eq > 0 constraints, are the [l,u] intervals simply recomputed bottom up and intersected with [-inf,0] or [0,inf] intervals at the constrained nodes? I f so, this should be stated clearly. The reason I am unclear is that it seems possible to derive stronger intervals b ased on the result of the linear programming solves (though I am not sure quite how to do this). If so those stronger intervals could be fed back into SLR to get better linear bounds. Low level comments: 0. What does SPEC mean? It's capitalized like an acryonym, but no definition is given. Is it just a strangely capitalized version of "specification"? Also, SPEC is a bad name for speed-related method like this given https://www.spec.org. 1. "Piece-wise" should be "piecewise". 2. There are a few spelling and grammar errors such as "Sepcifically", "we define safety property as that the confidence value of outputting targeted class", "The refinement runs iteratively by calling underlying efficient linear solver", etc. A proof reading pass is in order. 3. Line 213 is confusing, since it seems to imply that multiple overestimated nodes are split at once which I believe is false. I think this is likely just a plural vs. singular typo. 4. The inductive proof of property 1.2 in the supplementary material is needlessly complex, since it starts the induction at n = 1 (the first hidden layer) rather than n = 0 (the input). As a result, it gives essentially the same argument twice. 5. The proof of lemma 1.2 in the supplementary material can be replaced with a single line. It is obvious that once all the nodes are constrained to be linear that the result is perfectly approximable with linear functions. 6. The optimality of (1) on line 165 has a simpler, more intuitive proof than that given in supplementary material. For the upper bound, since relu is convex up the best approximating line touches the relu curve at both interval ends; this immediately gives the answer. Next, the optimal upper and lower bounds must have the same slope formula, since translating an upper bound down by its maximum error gives a lower bound with at most that same error and vice versa. 7. Although choosing linear interval bounds based on minimax error is simple, it's not obvious that it is the most efficient heuristic (for lower bounds, since the optimal upper bound is simultaneously optimal over the whole range by convexity). For example, if we somehow knew that for most of the subdivision tree a given relu node would be positive, we could choose a lower bound with higher slope and thus lower error in the positive part. There's no need to mention this observation if it isn't useful.

Reviewer 2



The paper presents a principled approach for formal verification of neural network properties by symbolic interval analysis of the network. The approach is made scalable through linear relaxation of the ReLU non linearity in the neural network. The upper and lower bounds for linear relaxation are computed using directed constraint refinement (where overestimated nodes are located and prioritized based on gradient magnitude, next they are split instead of relaxation to get tighter approximations and are solved with the safety constraints using an underlying linear solver) instead of naive interval propagation to make the approach scale to larger networks. Various safety properties such as robustness to adversarial inputs with L1, L2 and Linf norm and robustness to adversarial images with varying contrast and brightness are checked on ACAS Xu neural network, Drebin malware detection models, MNIST classifiers and NVIDIA self driving car model. The paper is clear and well written. The only problem I see with the submission is the novelty. The paper has combined methods from past literature but the experimental studies performed are strong. Update: The authors did a good job in their response and I would vote to accept it.

Reviewer 3



The authors present SPEC, a checker for formal safety analysis, in the form of norm-based constraints on the input-output, of ReLU-based neural networks. It does this by iteratively propagating relaxed bounds on the hidden nodes, looking for a counterexample, and tightening the bounds in case of a false positive. Propagating bounds is done by combining symbolic interval analysis with linear relaxation, which results in tighter bounds than symbolic analysis alone. Tightening the bounds is done using directed constraint refinement, which partitions the input domain in two around a the non-linear part of an overestimated node. The result is a checker that scales better with the number of hidden nodes in the network. Experiments cover different networks trained on different datasets and comparisons with other checkers. The paper is well-written, with a good and concise structure, although some of the notation could be made clearer (mainly not dropping the up & low subscripts). While the technique presented is not groundbreaking, as it combines several existing approaches without major modifications, the results do show the merits which is why I'm inclined to accept this paper. Experiments are performed on a range of datasets and models, though still medium-sized. Results mainly focus on the improved performance of SPEC compared to other approaches, checking one to several order of magnitudes faster. Some remarks: in the ablation studies, when removing the Directed Constraint Refinement, it is unclear where the timeout comes from (according to Figure 2). In what way is the result iteratively refined in this case (i.e. how are false positives handled?). In Figure 4b there is an increase in the number of cases SPEC can check for increased input bounds, which is counter-intuitive and not discussed. Finally, the others mention in the introduction that their method may help in training more robust networks and improve explainability, but don't discuss this further. Small comments: - Several typos here and there. - Incorrect table reference on p6. - Double-check formatting of references. ====== Update After reading the authors' rebuttal and the additional clarifications they will add to the paper, I've raised my score.